# Nowcasting GDP: An Application to Portugal

João B. Assunção [†] and Pedro Afonso Fernandes *,[†]

Católica Lisbon Forecasting Lab (NECEP), Católica Lisbon Research Unit in Business & Economics (CUBE), Católica Lisbon School of Business & Economics, Universidade Católica Portuguesa, Palma de Cima, Building 5, 1649-023 Lisboa, Portugal
* Correspondence: paf@ucp.pt; Tel.: +351-965-031-116
† These authors contributed equally to this work.

**Abstract:** Forecasting the state of an economy is important for policy makers and business leaders. When this is conducted in real-time, it is called nowcasting. In this paper, we present a method that shows how forecasting errors decline as additional contemporaneous information unfolds and becomes available. When the economic environment changes fast, as has happened often in the last decades across most developed economies, it is important to use forecasting methods that are both flexible and robust. This can be achieved with bridge equations and non-parametric estimates of the trend growth using only publicly available information. The method presented in this paper achieves, by the end of a quarter, an accuracy that is equivalent to the methods used by official entities.

**Keywords:** time series; macroeconomic forecasting; nowcasting; error correction models; combining forecasts

## 1. Introduction

Nowcasting is about estimating the present. In fact, it can be defined as assessing the current-quarter conditions [1] or current situation of the economy, rather than a pure forecast [2]. More precisely, *nowcasting* is the prediction of official statistical macroeconomic aggregates from quarterly accounts with the goal of providing governments, businesses and society in general with speedy quality assessments within a couple of days after the end of the quarter. Thus, nowcasts are produced when there are no official estimates of economic activity for the current or contemporaneous state of the economy. In fact, this paper is concerned with the prediction of macroeconomic aggregates that remain unavailable for rather long time spans, typically for 30 days or even 60 days after the end of the respective quarter.

Economic activity can be conceptualized as a stochastic process in continuous time. However, economic aggregates such as gross domestic product (GDP) are usually accumulated over three-month periods or quarterly, so they are a stochastic average of about 90 days of economic activity. All economies have idiosyncrasies that forecasters learn to appreciate over time. In the case of the Portuguese economy, net exports and tourism require extra statistical treatment to deal with seasonality for tracking the underlying state of economic activity. Policy makers should act quickly if they detect major deviations from historical trend growth, as was observed during the pandemic, the sovereign debt crisis or the Great Recession.

The importance of nowcasting goes beyond the purely predictive exercise. Obviously, governments, monetary authorities and companies are interested in knowing the current state of the economy and at the earliest time possible when only incomplete information is available [3] so that data-driven decisions can be made. In this scope, it is usually more important and interesting to assess economic dynamics and provide a qualitative assessment if things are broadly on track, or if a major change has occurred and it is observable in the high frequency data.

This paper describes the mixed-frequency method used by the NECEP–Católica Lisbon Forecasting Lab to predict the current state of an economy, applied to the Portuguese case with reference to the most relevant and swiftly available monthly series that can be regressed to estimate the quarterly GDP before the official figures. Here, we show the increasing accuracy of nowcasts as information about the current quarter unfolds. In particular, monthly data already available for the current quarter can be combined with lagged quarterly data to produce better nowcasts. Besides the scope of the results obtained for Portugal, the NECEP's method can be applied to every small, or even big, open economy due to its simplicity and low computational cost.

## 2. Literature Review

Estimation of current quarterly macroeconomic aggregates from higher-frequency data was introduced in the late 1960s by the German–American economist Otto Eckstein (1927–1984) at Data Resources, Inc., a company integrated in the meantime into IHS Markit Ltd. and now a part of S&P Global. However, it was Lawrence Robert Klein (1920–2013) who formulated the two main approaches to the problem [4].

On the one hand, an analyst can consider the main entries on the expenditure side of national accounts and then establish empirical bridge equations by aggregating high-frequency (monthly, weekly or daily) indicators into quarters and correlating those with quarterly components of the GDP. On the other hand, a forecaster can collect as much high-frequency data as are available in the current quarter and then extract the leading principal components of the quarterly averages of these multiple indicators and regress the GDP from them.

Bridge equation methods, also known as "tracking models" [5], explore the interrelationships between the expenditure (or income) components of GDP and readily available higher-frequency data. For example, the private consumption of durable goods might be highly correlated with car sales, and other relationships may be found for national quarterly account (NQA) entries such as gross fixed investments, inventory changes, government expenditure and net exports of goods and services [4].

Where possible, bridge equations are estimated from monthly data on both high-frequency indicators and NQA components. However, national accounts are not reported monthly in most cases, so quarterly bridge equations must be built by aggregating or averaging the monthly indicators into quarters. Typically, end-of-sample jagged values for those indicators are forecast with lower-frequency univariate models either to complete the current quarter or to forecast one or two quarters beyond.

The approach proposed by Klein and Sojo [4] builds separate bridge equations for nominal GDP components and price deflators, noting that real expenditure estimates can be obtained by dividing the estimated nominal value of each NQA component by an appropriate deflator, which may be highly correlated with consumer price index. Typically, a bridge equation relates each NQA entry to the quarterly averages of one or two highly correlated variables. Finally, a national accounting entity is used to sum up the estimated (real) NQA components into GDP.

Baffigi et al. [2], Diron [6] and Payne [7] are good examples of the application of bridge models (BMs). In the former case, bridge equations were estimated for aggregated GDP and components from the Euro area and three main countries (France, Germany and Italy) using as a benchmark univariate and multivariate statistical models and a small structural model; Baffigi et al. [2] show that national BMs perform better than their benchmark models, and that Euro-area GDP and its components are more precisely predicted by aggregating national forecasts. Again using bridge equations, Diron [6] evaluated three sources of forecast error, namely, model specification, unavailability of data in real-time and data revisions. Using a set of data vintages for the Euro area, she found that gains in accuracy of forecasts by using monthly data on actual activity rather than surveys and financial indicators are offset by the fact that the set of monthly data is harder to forecast and less timely than the latter set of data. Finally, Payne [7] used BMs to estimate US personal

consumption expenditures, residential and nonresidential investment, changes in private inventories, government expenditures, exports and imports, as well as to correct adding-up differences; he notes that improving component predictions does not always improve predictions of total GDP, and he lists several pitfalls to avoid in nowcasting exercises, including data changes and methodological revisions by national statistical entities.

The second approach to nowcasting is founded mainly on the contributions of Stock and Watson [8]. The premise of their models is that a few latent dynamic factors drive the comovements of a time series such as GDP, which is also affected by zero-mean idiosyncratic disturbances. These disturbances arise from measurement errors and from specific features of the data. The latent factors follow a time-series process, typically a vector autoregression (VAR). This kind of "medium data" or "data-rich" forecasting method without expenditure components [5] can cover more than 200 monthly macroeconomic indicators similar to the application of Giannone et al. [1] to the US economy.

In Portugal, the central bank (Banco de Portugal, BdP) has long experience in forecasting GDP using dynamic factor models (DFMs) with autoregressive components, the so-called diffusion index (DI) approach [8]. In this kind of model, a large number of predictors is summarized using a small number of indexes constructed by principal component analysis, and a dynamic factor model serves as the statistical framework for the estimation of the indexes and forecasting. As stressed by Dias et al. [9], these models require prior determination of the factors that reflect the top-ranked principal components; all other lower-ranked factors are entirely disregarded independent of their possible informational content. In order to avoid this limitation, Dias et al. [10] developed a targeting principle with a synthetic regressor that is computed as a linear combination of all the factors of the dataset weighted such that each factor takes into account both the relative size of the variance captured by it and its correlation with the variable of interest. Recent findings for Portuguese real GDP growth [11] suggest that this targeted diffusion index (TDI) approach could reduce the out-of-sample mean squared error (MSE) by 63% using a simple autoregressive model as the benchmark, while the gain with a DI model is about 50%. In this paper, we discuss the forecasting error of the TDI model, using it as benchmark for our bridge models.

Recently, a third approach to deal with data sampled at different frequencies has become "one of the main strands of the literature" [3]. Originally proposed by Ghysels et al. [12], MIxed DAta Sampling (MIDAS) models are essentially tightly parameterized, reduced-form regressions that involve processes sampled at different frequencies. While bridge equations regress lower-frequency data on aggregated higher-frequency data, MIDAS models can provide a direct forecast, namely, of quarterly GDP from parsimonious specifications based on distributed lag polynomials of monthly, weekly or even daily indicators. In this scope, MIDAS equations adopt parameterizations such as 'Exponential Almon Lag' or the Beta function with a relatively small number of hyperparameters estimated by nonlinear least squares (NLS) [3,13]. Ghysels et al. [12] showed that MIDAS regressions always lead to more efficient estimation than the typical approach of aggregating all series to the least-frequent sampling.

The combination of tracking (BM), data-rich (DFM) and mixed-frequency data (MIDAS) methods is possible. For example, bridge equations can include latent factors such as the Economic Climate Indicator provided by Statistics Portugal (INE) as regressors. The GDPNow model from the Federal Reserve Bank of Atlanta [5] is a good attempt in that direction, despite its complexity and detail level. A complete survey of forecasting models that deal with mixed-frequency data can be found in Foroni and Marcellino [3].

Here, we present an approach that combines bridge equations of the real GDP based on several covariates available on a monthly basis, including readily available common factors, with forecasts for the jagged missing values computed with a statistical filter. As described below, this multivariate approach performs as well as the TDI model developed by the Portuguese central bank (BdP), and slightly better than the univariate Theta method. Thus, this paper (1) describes the most-relevant aspects of the method developed by

NECEP to track the Portuguese economy and (2) shows the ability of this method to reduce forecasting errors as information related to the current quarter unfolds.

The key originality of the approach proposed here is the combination of structural and ad hoc non-parametric methods. In essence, the method replicates what may be used as tools by other forecasters and combines the various predictions with robust statistics. Significant tracking errors are dealt with by adding additional bridge equations to future forecasting exercises. In this paper, we show the out-of-sample performance of the method with the collection of models in use by the end of 2019.

## 3. Methodology

In general, our methodology is similar to the approach proposed by Miller and Chin [14]. Firstly, we use a model to predict each relevant seasonally adjusted monthly series $Z_{t:i}$ for the current quarter $t$, where $i = 1, 2, 3$ is the number of months of data available from it. When two months of data are available, the forecast for the third month of $t$ is

$$\hat{Z}_{t:3} \equiv Z_{t:2} + g_{t:2}, \tag{1}$$

where $g_{t:2} \equiv T_{t:2} - T_{t:1}$ is the first difference in the trend $T_{s:i}$ of the series $Z_{s:i}$, $s = 1, \ldots, t$, $i = 1, 2, 3$, which is chosen to minimize the sum of the square residuals $\varepsilon_{s:i} = Z_{s:i} - T_{s:i}$ and its weighted smoothness [15]:

$$\min\left\{\sum_{s,i} \varepsilon_{s:i}^2 + \lambda \sum_{s,i}[(T_{s:i} - T_{s:i-1}) - (T_{s:i-1} - T_{s:i-2})]^2\right\}, \tag{2}$$

where $\lambda$ is a penalty for the square of the second difference in the trend $T$. This application was conducted in levels, but it can also be performed in log-levels.

For particularly noisy raw series such as industrial production, cement sales or card transactions in point-of-sale (POS) or automated teller machines (ATM), we compute the moving average (MA) of the last three monthly observations, adding twice the first difference in the previous period's trend:

$$\tilde{Z}_{t:3} \equiv \frac{1}{3}(Z_{t:2} + Z_{t:1} + Z_{t-1:3}) + 2g_{t:1}. \tag{3}$$

The number of months $i$ of data available depends upon the time series and the day of the current quarter. For instance, at day 60, two months of the European Commission's Economic Sentiment Indicator (ESI) are available, but only the first month of industrial production data are ready to use; that is, this index is delayed one month. Table 1 indicates the selected series available at days 0, 30, 60 and 90 of the current quarter $t$, as well as at day 10 of the next quarter $t + 1$, here designated as day 100 for convenience.

**Table 1.** Number of months of data available at days 0, 30, 60, 90 and 100 of the current quarter $t$.

| Selected Time Series | Source | Day of the Current Quarter $t$ | | | | |
|---|---|---|---|---|---|---|
| | | 0 | 30 | 60 | 90 | 100 |
| Economic Sentiment Indicator (ESI) | EC | t−1:3 | t:1 | t:2 | t:3 | t:3 |
| Economic Climate Indicator (ICE) | INE | t−1:3 | t:1 | t:2 | t:3 | t:3 |
| Industrial Production Index | INE | t−1:2 | t−1:3 | t:1 | t:2 | t:2 |
| Cement sales | MF | t−1:2 | t−1:3 | t:1 | t:2 | t:3 |
| Car sales | ACAP | t−1:2 | t−1:3 | t:1 | t:2 | t:3 |
| Card transactions at POS/ATM | BdP | t−1:2 | t−1:3 | t:1 | t:2 | t:2 |
| Net exports of goods and services | BdP | t−1:1 | t−1:2 | t−1:3 | t:1 | t:1 |
| Net exports of goods | INE | t−1:1 | t−1:2 | t−1:3 | t:1 | t:2 |
| Consumer Price Index | INE | t−1:2 | t−1:3 | t:1 | t:2 | t:3 |
| Brent oil price | FRED | t−1:3 | t:1 | t:2 | t:3 | t:3 |
| Euro-coin | CEPR | t−1:3 | t:1 | t:2 | t:3 | t:3 |

As shown below, the quality of the nowcasting exercise improves to a great extent with the inclusion of information concerned with net exports. In fact, the monthly series of exports and imports of goods and services are delayed two months, with only one month available even 10 days after the end of the current quarter. However, two months of the series of net exports of goods (without services) are available at that time. Given that information, we perform a simple linear regression to forecast an additional figure for the net exports of goods and services to improve the accuracy of the nowcasts at day 100.

Secondly, we average the monthly data and the described forecasts for the jagged missing values in order to obtain quarterly figures. This procedure is necessary because our variable of interest (GDP) is issued quarterly, as previously explained. Thus, with two months of data available in quarter *t*, we compute:

$$\hat{Z}_t = \frac{1}{3}(Z_{t:1} + Z_{t:2} + \hat{Z}_{t:3}),$$ (4)

where $\hat{Z}_{t:3}$ can be replaced by $\tilde{Z}_{t:3}$ for noisy series. Net exports is still a special case in the sense that its quarterly value is estimated using a specific multiple regression model that takes into account as the dependent variable the net exports in volume provided by the NQA, available only 60 days after the end of the respective quarter, and, as independent variables, the monthly net exports of goods and services, the lagged price deflator and the Brent oil price. This procedure is required because the monthly series of net exports is provided in current prices, that is, in (nominal) values instead of (real) volumes.

Thirdly, the current quarter's year-over-year (y-o-y) GDP growth $y_t \equiv Y_t - Y_{t-4}$ is predicted using several bridge equations that explore different combinations of the monthly time series listed in Table 1. The first of these models, denoted by the superscript [1], includes, as independent variables, the growth accumulated in the last three quarters, that is, the difference in GDP level with a lag of three periods ($Y_{t-1} - Y_{t-4}$); the Economic Sentiment Indicator ($\hat{S}_t$), regularized by subtracting the historical average value (100); the Economic Climate Indicator ($\hat{E}_t$); and y-o-y differences in industrial production ($\hat{p}_t$) and cement sales ($\hat{c}_t$):

$$\hat{y}_t^{(1)} = \hat{\alpha}_0 + \hat{\alpha}_1(Y_{t-1} - Y_{t-4}) + \hat{\alpha}_2\hat{S}_t + \hat{\alpha}_3\hat{E}_t + \hat{\alpha}_4\hat{p}_t + \hat{\alpha}_5\hat{c}_t,$$ (5)

where $\hat{\alpha}_0, \ldots, \hat{\alpha}_5$ are the coefficients estimated with ordinary least squares (OLS).

The second model adds net exports to the independent variables already considered in Equation (5). The third and fourth models replace the Economic Climate Indicator ($\hat{E}_t$) with y-o-y changes in car sales and card transactions, respectively. Card transactions are expressed in real terms; that is, they were previously deflated with the consumer price index. The fifth model is based on the fourth but includes net exports. Finally, the sixth model replaces net exports with the Euro-coin real-time indicator of the Euro area economy, produced by CEPR and Banca d'Italia, a possible measure of external outlook.

For each model $j = 1, \ldots, 6$, we compute an alternative GDP growth estimate with a simple error correction mechanism that incorporates the last observed error $\epsilon_{t-1}^{(j)}$:

$$\hat{y}_t^{(jc)} = \hat{y}_t^{(j)} + \epsilon_{t-1}^{(j)} = \hat{y}_t^{(j)} + \left[y_{t-1} - \hat{y}_{t-1}^{(j)}\right].$$ (6)

Then, we use the median (X) of the twelve individual forecasts to provide a more robust estimation of GDP growth:

$$\hat{y}_t^{(c)} = X\left[\hat{y}_t^{(1)}, \hat{y}_t^{(1c)}, \ldots, \hat{y}_t^{(6)}, \hat{y}_t^{(6c)}\right].$$ (7)

As a benchmark, we calculate one- and two-period look-ahead forecasts using the Theta method proposed by Assimakopoulos and Nikolopoulos [16] and explained by

Hyndman and Billah [17]. Briefly, this method starts with the estimation of the following regression for the GDP level $Y$:

$$(1 - \theta)Y_t = \hat{a}_\theta + \hat{b}_\theta(t - 1), \tag{8}$$

for $\theta = 0$ and $\theta = 2$, where $t = 1, ..., n$ is the time index. Note that when $\theta = 0$, $\hat{a}_0$ and $\hat{b}_0$ are simply the parameters of the linear time trend fitted to the GDP quarterly series. Then, for each value of $\theta$, a new series $Y_{t,\theta}$ is constructed by:

$$Y_{t,\theta} = \hat{a}_\theta + \hat{b}_\theta(t - 1) + \theta Y_t. \tag{9}$$

The *h*-step ahead forecast is obtained by averaging the GDP forecasts for $\theta = 0, 2$:

$$\hat{Y}_t(h) = \frac{1}{2}\left[\hat{Y}_{t,0}(h) + \hat{Y}_{t,2}(h)\right], \tag{10}$$

where $\hat{Y}_{t,0}(h)$ is obtained by extrapolating the linear time trend:

$$\widehat{Y}_{t,0}(h) = \hat{a}_0 + \hat{b}_0(t + h - 1), \tag{11}$$

and $\hat{Y}_{t,2}(h)$ results from simple exponential smoothing (SES) on series $\{Y_{t,2}\}$:

$$\widehat{Y}_{t,2}(h) = \gamma Y_{t,2} + (1 - \gamma)\hat{Y}_{t-1,2}, \tag{12}$$

where the starting value $\hat{Y}_{1,2} = Y_{1,2}$ and the smoothing parameter $\gamma = 0.3$. In a classic paper, Muth [18] found that the exponential smoothing or 'adaptive expectations' forecasting scheme is optimal for a first-order moving-average stochastic process, see also Hamilton [19] and Ljungqvist and Sargent [20]. Thus, we assume that the smoothing parameter $\gamma$ is (approximately) equal to the moving average coefficient of the Portuguese real GDP, estimated in first-differences from quarterly data (1995Q1–2019Q4), which is $0.28 \approx 0.3$.

## 4. Data

In this application, we considered the y-o-y differences in real GDP provided by Statistics Portugal (INE) since the first quarter of 1996 till the fourth quarter of 2019 (total of 96 observations), available 45 days after the end of the respective quarter, complemented by monthly data concerned with industrial production, cement sales, car sales, card transactions, net exports, prices and coincident indicators or common factors (ESI, ICE and Euro-coin). The most-recent data were not considered in this paper because the COVID-19 pandemic and lockdowns created a structural break in the GDP series, namely, for Portugal in the first quarter of 2020, that required a different forecasting strategy and method.

An out-of-sample rolling forecasting exercise was performed from the first quarter of 2002 to assess the relative performance of the described nowcasting models. Each current-quarter growth was estimated with the data available at days 0, 30, 60, 90 and 100. For days 0 and 30, the benchmark is the two-steps-ahead forecast given by the Theta method, recalling that GDP change was available only at day 45 of the current quarter (and net exports at day 60) before 2020. For days 60, 90 and 100, we used the one-period look-ahead Theta forecast as benchmark.

## 5. Results

The results obtained can be summarized in three points. Firstly, the estimated bridge equations have high quality from a statistical point of view; that is, their design favors adherence to the data. Secondly, the difference between the GDP growth estimated by these models and the observed data, that is, the forecast error or deviance, becomes smaller as information about the current quarter unfolds; thus, high-frequency data such as industrial production, car sales and net exports may contain relevant information to track GDP

evolution "just-in-time" and right before the publishing of the official figures. Thirdly, the economic implication of our work is that policy makers (and managers) should track these monthly indicators in order to make strategic and effective decisions that influence or depend upon economic growth.

The estimated coefficients for the full sample (1996Q1–2019Q4) are presented in Table A1. In general, these coefficients are 1% or 5% statistically significant through the six models; the few exceptions include the industrial production index. Particularly relevant is the coefficient associated with the cumulative growth (in the last three quarters) of real GDP, suggesting the auto-correlated nature of the dependent variable. The six models have high, significant F statistics and adjusted $R^2$ above 90%, specifically Models 2 and 5 with net exports.

Truncating the data may have a limited impact on these estimates. In fact, the coefficients presented in Table A5 for roughly half of the sample (40 observations from 1996Q1 to 2005Q4) are similar to those condensed in Table A1. Nevertheless, industrial production, and sometimes the Economic Sentiment Indicator (ESI), change in exports y-o-y and the CEPR's Euro-coin indicator, had decreased significance in all models.

Figure 1 illustrates the out-of-sample mean absolute error (MAE) for the first model, progressively updated with more data over the current quarter. For each quarter from 2002Q1 to 2019Q4, we computed the direct (simple) forecast without last-error correction, the forecast with last-error correction (as described previously) and the median of these two estimates.

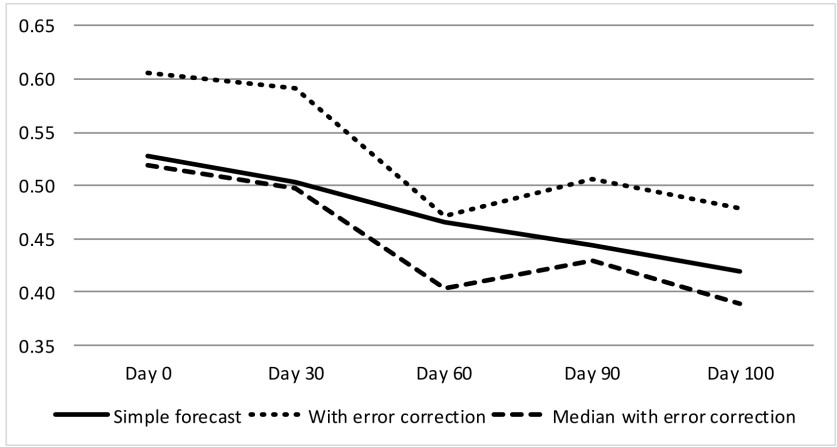

**Figure 1.** Mean absolute out-of-sample forecast error of Model 1 with and without last-error correction and progressively updated with more data over the current quarter (percentage points, 2002Q1–2019Q4).

A first observation suggests that the simple mechanism of picking the last out-of-sample error might not be effective for reducing the MAE. Nevertheless, the intermediate point between the simple and corrected forecasts always performs better. This is an expected result in the sense that the median has proven very powerful for attenuating or even removing noise in time series [21].

A second observation is that the MAE becomes smaller from day 0 to day 100; that is, the incorporation of more information related to the current quarter improves the quality of the nowcasting exercise. This result is observed in all models, and it is particularly evident from day 30 to day 60; recall that national accounts (GDP and net exports) for the previous quarter became fully available only at day 60 after the end of the respective quarter. Thus, that information should be extremely important to improve the accuracy of GDP growth estimates in addition to the readily available data. As far as Model 1 is concerned, the MAE increases slightly from day 60 to day 90, but this is not observed in Models 2 to 5, where the error decreases monotonically.

Figures 2 and 3 present the cumulative absolute out-of-sample error of the median of the simple forecasts given by the six models without and with error correction, respectively. The reduction in the absolute error by including the last error correction in the median is evident especially for day 60 and beyond. Additionally, both figures confirm the relevance of using, progressively, more and more data as the current quarter unfolds. In fact, monthly data are still important in the sense that the cumulative absolute error decreases from day 0 to day 30 and even more from day 60 to days 90 and 100. Our forecasts are typically published at day 100 with a sight error reduction compared to day 90 due to fresh data concerned with the net export of goods, car sales and cement sales.

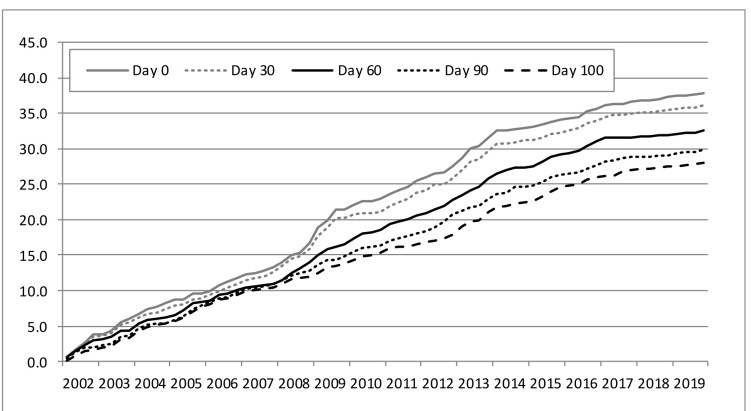

**Figure 2.** Cumulative absolute out-of-sample forecast error of the median of Models 1 to 6 progressively updated with more data over the current quarter (percentage points, 2002Q1–2019Q4).

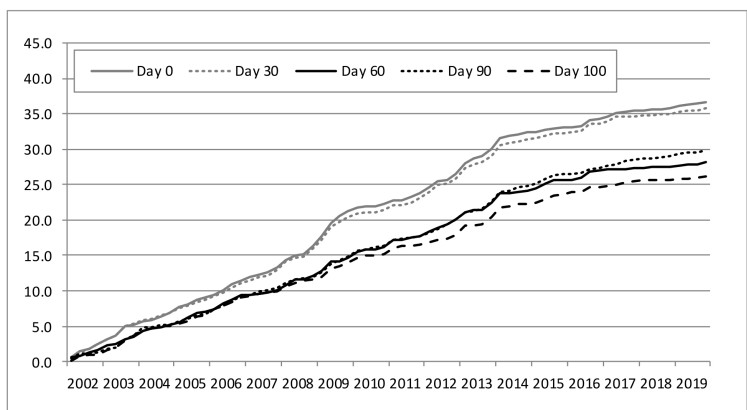

**Figure 3.** Cumulative absolute out-of-sample forecast error of the median of Models 1 to 6 with last-error correction progressively updated with more data over the current quarter (percentage points, 2002Q1–2019Q4).

The gain in terms of out-of-sample MAE between day 0 and day 60 is 0.08 percentage points (pp) from 0.53 to 0.45, as suggested by the last column of Table A2. With last-error correction, the gain is slightly better, 0.09 pp, from 0.51 to 0.42; see Table A3. The inclusion of more high-frequency data related to the current quarter can improve the MAE another 0.06 pp, for a final mark of 0.36 for day 100 with last-error correction. This kind of improvement is also visible in mean squared error (MSE) and its root, which is directly comparable with MAE.

As suggested by Figures 4–7, our approach performs quite better than the Theta method. We also found that last-error correction may increase the MAE of the estimates computed with that method. Thus, error correction mechanisms such as the one employed here should be avoided or used with care as far as the Theta method is concerned.

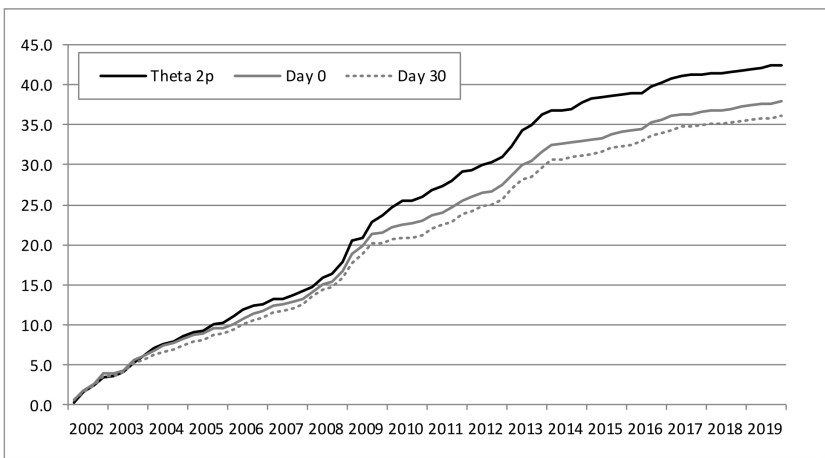

**Figure 4.** Cumulative absolute out-of-sample forecast error of the median of Models 1 to 6 given the data available at days 0 and 30 of the current quarter compared with the two-step-ahead Theta forecast (percentage points, 2002Q1–2019Q4).

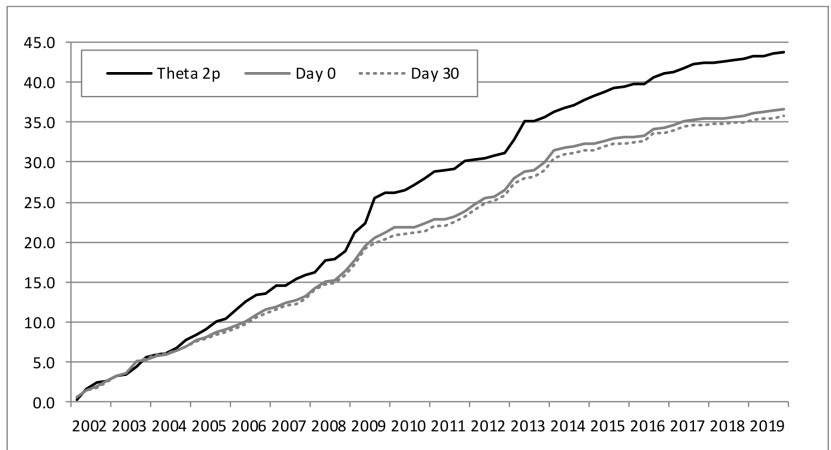

**Figure 5.** Cumulative absolute out-of-sample forecast error of the median of Models 1 to 6 with last-error correction given the data available at days 0 and 30 of the current quarter compared with the two-step-ahead Theta forecast (percentage points, 2002Q1–2019Q4).

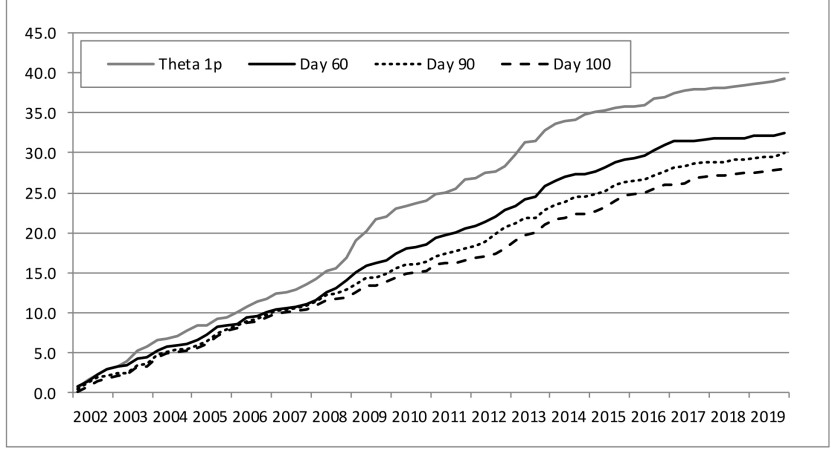

**Figure 6.** Cumulative absolute out-of-sample forecast error of the median of Models 1 to 6 given the data available at days 60, 90 and 100 of the current quarter compared with the one-step-ahead Theta forecast (percentage points, 2002Q1–2019Q4).

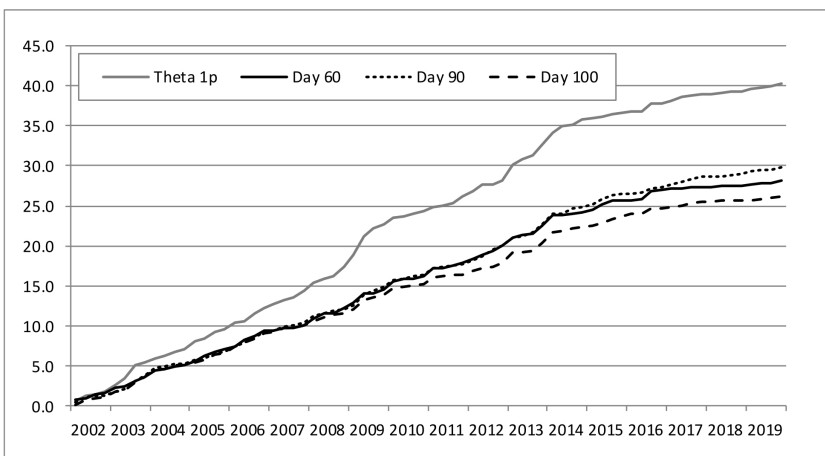

**Figure 7.** Cumulative absolute out-of-sample forecast error of the median of Models 1 to 6 with last-error correction given the data available at days 60, 90 and 100 of the current quarter compared with the one-step-ahead Theta forecast (percentage points, 2002Q1–2019Q4).

In short, the key feature of our method is the combination of estimates from different models using robust statistics. This procedure is quite flexible because we can add new models every time a relevant variable emerges in the short-run. For example, we introduced retail trade and fuel sales in the pool of models during the pandemic. This flexibility proves to be very important to maintain the quality of our models or even to further reduce the forecasting error.

## 6. Discussion

The targeted diffusion index (TDI) model estimated by Dias et al. [11] used a comparable dataset but for a shorter period (2002Q1–2015Q4). For this sub-sample, our approach with last-error correction gave an MAE of 0.42, which is close to the 0.41 from the TDI model (see Table A4). In root MSE, the difference between the two methods is insignificant at days 60 and 90, and our approach may perform better at day 100, when the NECEP forecasts are regularly published.

Thus, the method described in this paper can perform as well as the standard models used by central banks and better than the Theta method in terms of out-of-sample errors, as described. In addition, our method is simpler and more transparent than dynamic factor models (DFM). For example, Giannone et al. [1] applied the Kalman filter, a rather complex technique, to more than 200 time series, while we simply linearly regress GDP on about ten series. Naturally, our information set is smaller, so we may lose key data that could reduce the forecasting error further.

Additionally, our bridge models are simply confined to GDP as the dependent variable, while Payne [7], Baffigi et al. [2] and Higgins [5] have separate models for the expenditure components of GDP. Thus, we lose the possibility of having partial estimates, namely for private consumption and investment, in order to keep things simple and flexible.

Nevertheless, we are not obsessed with the simplicity or effectiveness of our method. Most important is the trade-off between flexibility and robustness. On the one hand, about 20 years of nowcasting practice in NECEP tells us that it is very important to construct "just-in-time" models using newly relevant high-frequency indicators. On the other hand, simple, robust techniques such as combining different estimates using the median operator have proven quite effective.

In any case, by averaging higher-frequency (monthly) data on a lower (quarterly) frequency, a lot of potentially useful information might be destroyed [3]. Thus, direct modeling of different frequencies, namely by using MIDAS regression models [12], ought to be pondered in a future development of this research.

## 7. Conclusions

This paper shows how to use robust statistical models to nowcast macroeconomic variables of interest, such as GDP, based on readily available qualitative or quantitative high-frequency (monthly) indicators. In the former case, we have coincident indicators of business sentiment or economic climate obtained from surveys; in the second case, we have quantitative indicators such as industrial production, cement and car sales, or the value of net exports. Both types of indicators can be aggregated quarterly by averaging and can be used in simple multivariate linear regressions of GDP growth. In this aggregation, the monthly jagged values, not yet available for the current quarter, can be estimated using a simple statistical filter, correcting any excessive noise with moving averages and using the median to combine the forecasts, and eventually correcting using the last observed forecast error. In fact, the originality of our approach results from the combination of structural bridge equations with ad hoc non-parametric methods and robust statistics.

The key insight is that we can significantly reduce the out-of-sample mean absolute error (MAE) as the information about the current quarter unfolds. At the beginning of the current quarter (day 0), the MAE to forecast GDP growth is still 0.51 percentage points. Noting that the average GDP growth observed in Portugal is close to 0.5%, that error means that we cannot infer whether growth is going to occur or not.

Nevertheless, when the NECEP's quarterly newsletter is published (10 days after the end of the current quarter), the error becomes smaller, about 0.36 percentage points. Thus, with the statistical procedures described above, we can produce robust estimates of GDP growth with a good confidence level about 35 days before the official flash estimate, which was published 45 days after the end of the quarter of interest until 2020.

In fact, these procedures can lead to predictions with an average error similar to that of the targeted diffusion index (TDI) adopted by the Portuguese Central Bank (BdP), which may involve the treatment of tens or even hundreds of time series. The method described throughout this paper has proved effective against the recognized Theta method, and replication in other countries besides Portugal is straightforward due to its simplicity and low computational cost.

Future developments of this research may involve estimation of dynamic factor models (DFM) following the contributions of Giannone et al. [1] and Azevedo et al. [22]. First observations have already been made regarding using the Kalman filter in order to obtain a common factor that can be used as a predictor in the described bridge equations, eventually replacing the Economic Sentiment Indicator (ESI) or the Economic Climate Indicator (ICE). Another line of research could cover regularized regression methods (e.g., LASSO), regression trees, random forests, neural networks, reinforcement learning and other machine-learning techniques, following the seminal contributions of Mullainathan and Spiess [23] and Dauphin et al. Dauphin et al. [24]. Finally, MIDAS models [12] can be estimated using different parameterizations and lags.

**Author Contributions:** Conceptualization, J.B.A.; methodology, J.B.A.; software, P.A.F.; validation, J.B.A.; formal analysis, J.B.A. and P.A.F.; investigation, J.B.A. and P.A.F.; resources, J.B.A. and P.A.F.; data curation, P.A.F.; writing—original draft preparation, P.A.F.; writing—review and editing, J.B.A. and P.A.F.; visualization, P.A.F.; supervision, J.B.A.; project administration, J.B.A.; funding acquisition, J.B.A. All authors have read and agreed to the published version of the manuscript.

**Funding:** This research was supported by Fundação para a Ciência e Tecnologia (FCT), Lisbon, Portugal, under a post-doctoral grant with reference CUBE-MACROECO-BPD4 from Católica Lisbon Research Unit in Business & Economics (UID/GES/00407/2020), and by the research project PTDC/EGE-ECO/27884/2017 from the same unit.

**Institutional Review Board Statement:** Not applicable.

**Informed Consent Statement:** Not applicable.

**Data Availability Statement:** Data and code can be provided by request.

**Conflicts of Interest:** The authors declare no conflict of interest.

## Abbreviations

The following abbreviations are used in this manuscript:

| | |
|---|---|
| ACAP | Portuguese Automotive Association |
| ATM | Automated Teller Machines |
| BdP | Banco de Portugal |
| BM | Bridge Models |
| CEPR | Centre for Economic Policy Research |
| DFM | Dynamic Factor Models |
| DI | Diffusion Index |
| EC | European Commission |
| ESI | Economic Sentiment Indicator |
| FCT | Portuguese Foundation for Science and Technology |
| FRED | Federal Reserve Bank of Atlanta, USA |
| GDP | Gross Domestic Product |
| HP | Hodrick and Prescott |
| ICE | Economic Climate Indicator |
| INE | Statistics Portugal |
| LASSO | Least Absolute Shrinkage and Selection Operator |
| MA | Moving Average |
| MAE | Mean Absolute Error |
| MIDAS | MIxed DAta Sampling regression models |
| MSE | Mean Squared Error |
| NQA | National Quarterly Accounts |
| NLS | Nonlinear Least Squares |
| OLS | Ordinary Least Squares |
| POS | Point of Sales |
| pp | Percentage points |
| SES | Simple Exponential Smoothing |
| TDI | Targeted Diffusion Index |
| US | United States of America |
| VAR | Vector Autoregressions |
| y-o-y | year-over-year |

## Appendix A

**Table A1.** Estimated coefficients (full sample 1996Q1–2019Q4).

| Coefficients | Model | | | | | |
|---|---|---|---|---|---|---|
| | 1 | 2 | 3 | 4 | 5 | 6 |
| Cum. growth ($Y_{t-1} - Y_{t-4}$) | 0.497 *** | 0.477 *** | 0.545 *** | 0.458 *** | 0.448 *** | 0.525 *** |
| | (0.080) | (0.064) | (0.087) | (0.080) | (0.069) | (0.080) |
| Econ. Sentiment Indic. ($\hat{S}_t$) | 0.055 *** | 0.020 | 0.110 *** | 0.108 *** | 0.088 *** | 0.039 ** |
| | (0.016) | (0.014) | (0.013) | (0.011) | (0.011) | (0.018) |
| Econ. Climate Indic. ($\hat{E}_t$) | 0.310 *** | 0.366 *** | | | | 0.312 *** |
| | (0.060) | (0.050) | | | | (0.059) |
| Industrial production ($\hat{p}_t$) | 0.068 *** | 0.015 | 0.069 *** | 0.076 *** | 0.034 ** | 0.055 *** |
| | (0.016) | (0.015) | (0.018) | (0.016) | (0.015) | (0.017) |
| Cement sales ($\hat{c}_t$) | 0.018 *** | 0.015 *** | 0.025 *** | 0.019 *** | 0.017 *** | 0.018 *** |
| | (0.005) | (0.004) | (0.005) | (0.005) | (0.004) | (0.005) |
| Car sales | | | 0.010 ** | | | |
| | | | (0.004) | | | |

**Table A1.** *Cont.*

| Coefficients | Model | | | | | |
|---|---|---|---|---|---|---|
| | 1 | 2 | 3 | 4 | 5 | 6 |
| Card transactions | | | | 0.071 *** | 0.071 *** | |
| | | | | (0.013) | (0.012) | |
| Exports | | 0.042 *** | | | 0.037 *** | |
| | | (0.013) | | | (0.014) | |
| Imports | | 0.063 *** | | | 0.051 *** | |
| | | (0.014) | | | (0.015) | |
| Euro-coin | | | | | | 0.662 *** |
| | | | | | | (0.107) |
| Constant | 0.736 *** | 0.269 | 0.937 *** | 0.457 *** | 0.095 | 0.381 *** |
| | (0.092) | (0.098) | (0.090) | (0.122) | (0.124) | (0.132) |
| Observations | 96 | 96 | 96 | 96 | 96 | 96 |
| $R^2$ | 0.948 | 0.968 | 0.938 | 0.950 | 0.964 | 0.951 |
| Adjusted $R^2$ | 0.946 | 0.965 | 0.934 | 0.947 | 0.961 | 0.947 |
| Residual Std. Error | 0.556 | 0.443 | 0.611 | 0.549 | 0.472 | 0.547 |
| F statistic | 331.1 *** | 379.8 *** | 271.7 *** | 339.7 *** | 333.9 *** | 285.9 *** |

*p*-values: ** $p < 0.05$; *** $p < 0.01$.

**Table A2.** Mean squared and absolute out-of-sample forecast errors of the median of Models (1) to (6) conditioned with available data over the current quarter (full sample 2002Q1–2019Q4).

| Data Available at: | MSE | Root MSE | MAE |
|---|---|---|---|
| Day 0 | 0.45 | 0.67 | 0.53 |
| Day 30 | 0.40 | 0.63 | 0.50 |
| Benchmark: Theta model 2p | 0.60 | 0.77 | 0.59 |
| Day 60 | 0.30 | 0.54 | 0.45 |
| Day 90 | 0.25 | 0.50 | 0.42 |
| Day 100 | 0.23 | 0.48 | 0.39 |
| Benchmark: Theta model 1p | 0.49 | 0.70 | 0.54 |

**Table A3.** Mean squared and absolute out-of-sample forecast errors of the median of Models (1) to (6) conditioned with available data over the current quarter after last-error correction (full sample 2002Q1–2019Q4).

| Data Available at: | MSE | Root MSE | MAE |
|---|---|---|---|
| Day 0 | 0.42 | 0.65 | 0.51 |
| Day 30 | 0.41 | 0.64 | 0.50 |
| Benchmark: Theta model 2p | 0.69 | 0.83 | 0.61 |
| Day 60 | 0.26 | 0.51 | 0.42 |
| Day 90 | 0.26 | 0.51 | 0.42 |
| Day 100 | 0.23 | 0.48 | 0.36 |
| Benchmark: Theta model 1p | 0.52 | 0.72 | 0.56 |

**Table A4.** Mean squared and absolute out-of-sample forecast errors of the median of Models (1) to (6) conditioned with available data over the current quarter after last-error correction (sub-sample 2002Q1–2015Q4).

| Data Available at: | MSE | Root MSE | MAE |
|---|---|---|---|
| Day 0 | 0.52 | 0.72 | 0.59 |
| Day 30 | 0.49 | 0.70 | 0.58 |
| Benchmark: Theta model 2p | 0.85 | 0.92 | 0.70 |
| Day 60 | 0.32 | 0.56 | 0.46 |
| Day 90 | 0.32 | 0.56 | 0.47 |
| Day 100 | 0.28 | 0.53 | 0.42 |
| Benchmark: Theta model 1p | 0.64 | 0.80 | 0.66 |
| Benchmark: TDI model | 0.30 | 0.55 | 0.41 |

**Table A5.** Estimated coefficients (sub-sample 1996Q1–2005Q4).

| Coefficients | Model | | | | | |
|---|---|---|---|---|---|---|
| | **1** | **2** | **3** | **4** | **5** | **6** |
| Cum. growth ($Y_{t-1} - Y_{t-4}$) | 0.512 *** | 0.470 *** | 0.503 *** | 0.561 *** | 0.550 *** | 0.522 *** |
| | (0.167) | (0.140) | (0.157) | (0.169) | (0.156) | (0.170) |
| Econ. Sentiment Indic. ($\hat{S}_t$) | 0.029 | -0.007 | 0.090 *** | 0.091 *** | 0.058 ** | 0.020 |
| | (0.030) | (0.027) | (0.020) | (0.022) | (0.024) | (0.036) |
| Econ. Climate Indic. ($\hat{E}_t$) | 0.388 *** | 0.375 *** | | | | 0.398 *** |
| | (0.123) | (0.108) | | | | (0.125) |
| Industrial production ($\hat{p}_t$) | 0.023 | 0.029 | 0.032 | 0.040 | 0.049 | 0.027 |
| | (0.032) | (0.027) | (0.029) | (0.032) | (0.030) | (0.034) |
| Cement sales ($\hat{c}_t$) | 0.018 *** | 0.015 ** | 0.023 *** | 0.019 *** | 0.020 *** | 0.018 *** |
| | (0.006) | (0.006) | (0.005) | (0.006) | (0.006) | (0.006) |
| Car sales | | | 0.031 *** | | | |
| | | | (0.008) | | | |
| Card transactions | | | | 0.066 ** | 0.038 | |
| | | | | (0.024) | (0.025) | |
| Exports | | 0.020 | | | 0.035 | |
| | | (0.031) | | | (0.036) | |
| Imports | | 0.092 *** | | | 0.073 ** | |
| | | (0.025) | | | (0.029) | |
| Euro-coin | | | | | | 0.200 |
| | | | | | | (0.463) |
| Constant | 0.6632 *** | 0.302 | 1.348 *** | 0.465 | 0.393 | 0.569 |
| | (0.233) | (0.216) | (0.179) | (0.307) | (0.284) | (0.321) |
| Observations | 40 | 40 | 40 | 40 | 40 | 40 |
| $R^2$ | 0.918 | 0.946 | 0.927 | 0.913 | 0.931 | 0.919 |
| Adjusted $R^2$ | 0.906 | 0.934 | 0.916 | 0.900 | 0.915 | 0.904 |
| Residual Std. Error | 0.594 | 0.499 | 0.562 | 0.613 | 0.565 | 0.601 |
| F statistic | 76.5 *** | 79.9 *** | 82.2 *** | 71.6 *** | 62.2 *** | 62.3 *** |

*p*-values: ** $p < 0.05$; *** $p < 0.01$.

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
