# Peer review of "Nowcasting GDP: An Application to Portugal"

_forecasting, doi:10.3390/forecast4030039_

Round 1

Reviewer 1 Report

Dear Authors,

The work is too technical. You are forecasting specific indicators for Portugal, so it would be necessary to present the results properly (the economic implications). 

Author Response

Many thanks for your peer-review. Following your recommendations, the second version of the manuscript “Nowcasting GDP: an application to Portugal” was updated and revised in the following fashion:

  • The abstract was rewritten and now is more focused on the relevant issue, describing briefly the purpose, method used, its relative performance and main findings;
  • The introduction was revised and updated in order to give a precise definition of nowcasting and to stress its purpose and economic implications in a wide and international perspective, that is, not confined to the (Portuguese) case developed in the paper;
  • The literature review was enriched with new references concerned with bridge models, namely, Baffigi et al. (International Journal of Forecasting, 2004), Diron (Journal of Forecasting, 2008) and Payne (Business Economics, 2000), with a brief description of the methods and main findings in each case; additionally, a new paragraph concerned with MIxed DAta Sampling (MIDAS) regression models was included, following the original contribution of Ghysels et al. (2004);
  • The literature review’s paragraph concerned with dynamic factor models (DI/TDI) was condensed and the (complementary) reference Bay and Ng (2008) was removed, noting that the present paper is concerned with a different method (bridge equations), besides the discussion of the results using a TDI model (from Banco de Portugal) as benchmark;
  • A discussion section was inserted with, namely, the material concerned with the comparative analysis with the TDI model’s results from Banco de Portugal (Dias et al., 2016) and the trade-off between flexibility and robustness; the main limitation of the study is also pointed out;
  • The conclusion was updated, namely, with a list of future research developments (dynamic factor models, machine learning techniques and MIDAS models) and miscellaneous remarks.

Reviewer 2 Report

Dear authors, your paper is devoted to an interesting and, in the current conditions, also a very important issue.

I do like and appreciate the method used as well as the presented calculation, which could be very useful not only in the Portuguese environment. However, there are some weak points in the manuscript which should be revised and improved:

1. the abstract of the paper is not attractive and it does not cover the relevant issue. Please, rewrite the abstract following the instructions of the journal and mention the background of the paper, the main purpose, methods used and crucial findings & value added.

2. please, separate the introduction and literature review sections.

3. in the introduction section, please, add the main aim of the study, its originality and its purpose. Do not forget to highlight why the one-country study is of interest to an international audience. 

4. in the literature review section, please, add more Q1/Q2 WoS/ Scopus papers. In the current version of the paper, there are only 13 references used, which is not sufficient for a scientific paper. 

5. the discussion section is missing. It is necessary to discuss/compare your findings in the context of other relevant studies published worldwide. 

6. it is also appropriate to add more information about the data used in the analyses, lines 170 etc.,  e.g. some basic descriptions. 

7. the conclusion section is very weak. It is recommended to broaden the section devoted to the crucial findings and highlight the practical implications and theoretical contributions of the study. Please, be more accurate about the study limitations and future research challenges. 

Author Response

Many thanks for your peer-review. Following your recommendations, the second version of the manuscript “Nowcasting GDP: an application to Portugal” was updated and revised in the following fashion:

  • The abstract was rewritten and now is more focused on the relevant issue, describing briefly the purpose, method used, its relative performance and main findings;
  • The former introduction was divided in two sections: introduction (properly) and literature review;
  • The introduction was revised and updated in order to give a precise definition of nowcasting and to stress its purpose and economic implications in a wide and international perspective, that is, not confined to the (Portuguese) case developed in the paper;
  • The literature review was enriched with new references concerned with bridge models, namely, Baffigi et al. (International Journal of Forecasting, 2004), Diron (Journal of Forecasting, 2008) and Payne (Business Economics, 2000), with a brief description of the methods and main findings in each case; additionally, a new paragraph concerned with MIxed DAta Sampling (MIDAS) regression models was included, following the original contribution of Ghysels et al. (2004);
  • The literature review’s paragraph concerned with dynamic factor models (DI/TDI) was condensed and the (complementary) reference Bay and Ng (2008) was removed, noting that the present paper is concerned with a different method (bridge equations), besides the discussion of the results using a TDI model (from Banco de Portugal) as benchmark;
  • The originality/value-added of the paper was stressed in a new, final paragraph in the literature review, as well in the conclusion;
  • A discussion section was inserted with, namely, the material concerned with the comparative analysis with the TDI model’s results from Banco de Portugal (Dias et al., 2016) and the trade-off between flexibility and robustness; the main limitation of the study is also pointed out;
  • A basic description of the data was inserted at end of the first paragraph in the results section;
  • The conclusion was updated, namely, with a list of future research developments (dynamic factor models, machine learning techniques and MIDAS models) and miscellaneous remarks;
  • The total number of references increases from 13 to 24 (+11).

Reviewer 3 Report

Please see ttached PDF

Author Response

Many thanks for your peer-review. Following your recommendations, the second version of the manuscript “Nowcasting GDP: an application to Portugal” was updated and revised in the following fashion:

  • The abstract was rewritten and now is more focused on the relevant issue, describing briefly the purpose, method used, its relative performance and main findings;
  • The introduction was revised and updated in order to give a precise definition of nowcasting and to stress its purpose and economic implications in a wide and international perspective, that is, not confined to the (Portuguese) case developed in the paper;
  • In addition, the introduction now clarifies that “this paper is concerned with the prediction of macroeconomic aggregates that remain unavailable for rather long-time spans, typically for 30 days or even 60 days after the end of the respective quarter”;
  • The literature review was enriched with new references concerned with bridge models, namely, Baffigi et al. (International Journal of Forecasting, 2004), Diron (Journal of Forecasting, 2008) and Payne (Business Economics, 2000), with a brief description of the methods and main findings in each case; additionally, a new paragraph concerned with MIxed DAta Sampling (MIDAS) regression models was included, following the original contribution of Ghysels et al. (2004);
  • A discussion section was inserted with, namely, the material concerned with the comparative analysis with the TDI model’s results from Banco de Portugal (Dias et al., 2016) and the trade-off between flexibility and robustness; the main limitation of the study is also pointed out;
  • The conclusion was updated, namely, with a list of future research developments (dynamic factor models, machine learning techniques and MIDAS models) and miscellaneous remarks;
  • The total number of references increases from 13 to 24 (+11);
  • An exhaustive revision of the English was made, namely, to correct the no concordance between subjects and verbs (singular versus plural) and the typing errors.

Reviewer 4 Report

1. In the introduction section, please explain why your method can combine the data-rich method such as dynamic factor model. In particular, you state "bridge equations can include latent factors", but I don't see you did.

2. In Tables A1 and A4, the abbreviations of coefficients such as "ESI-100" and "ICE" are hard to link to the predictors in model (5). 

3. A word why you chose SES parameter 0.3 is appreciated

4. In English, singular and plural are mixed in several places. For example, "these models requires" (page 2), "The third and fourth models replaces" (page 4), "whether growth are" (page 9) etc.

Author Response

Many thanks for your peer-review. Following your recommendations, the second version of the manuscript “Nowcasting GDP: an application to Portugal” was updated and revised in the following fashion:

  • The abstract was rewritten and now is more focused on the relevant issue, describing briefly the purpose, method used, its relative performance and main findings;
  • The introduction was revised and updated in order to give a precise definition of nowcasting and to stress its purpose and economic implications in a wide and international perspective, that is, not confined to the (Portuguese) case developed in the paper;
  • Now, the paper clarifies that “bridge equations can include latent factors like the Economic Climate Indicator provided by Statistics Portugal (INE)”;
  • At the end of the methodology, a new footnote was introduced to explain the value of the parameter of exponential smoothing adopted in the scope of the Theta method;
  • A discussion section was inserted with, namely, the material concerned with the comparative analysis with the TDI model’s results from Banco de Portugal (Dias et al., 2016) and the trade-off between flexibility and robustness; the main limitation of the study is also pointed out;
  • The conclusion was updated, namely, with a list of future research developments (dynamic factor models, machine learning techniques and MIDAS models) and miscellaneous remarks;
  • The abbreviations/legends of the coefficients in tables A.1 and A.4 were revised;
  • An exhaustive revision of the English was made, namely, to correct the no concordance between subjects and verbs (singular versus plural) and the typing errors.

Round 2

Reviewer 1 Report

The article is not in a proper form. The authors did not respond to the requirements. 

Author Response

Many thanks. 

Following your comments that our article is not in a proper form and it is too technical, so it would be necessary to present the results properly with its economic implications, we started by dividing the previous Results section in Data and Results properly. Then, we wrote two additional paragraphs, one at the beginning of the Results section, the other at the end of the same section:

The results obtained can be summarised in three points. Firstly, the estimated bridge equations have a high quality from the statistical point of view, that is, their design favours the adherence to the data. Secondly, the difference between the GDP growth estimated by these models and the observed data, that is, the forecast error or deviance becomes smaller and smaller as information about the current quarter unfolds; thus, high frequency data like industrial production, car sales or net exports may contain relevant information to track the GDP evolution “just-in-time” and right before the publishing of the official figures. Thirdly, the economic implication of our work is that the policy-makers (and managers) should track these monthly indicators in order to take strategic and effective decisions that influence or depend upon the economic growth.

(...)

In short, the key feature of our method is the combination of estimates from different models using robust statistics. This procedure is quite flexible because we can add new models every time a new, relevant variable emerge in the short-run. For example, we introduced retail trade and fuel sales in the pool of models during the pandemic. This flexibility proved to be very important to keep the quality of our models or even to reduce the forecasting errors further.

Reviewer 2 Report

Dear authors, thank you very much for the improvements. Please, try to improve the discussion section, giving the arguments (pros and cons) in the context of other studies.

Author Response

Many thanks.

Following your last comments, we developed the Discussion section with the pros and cons in the context of other studies. In this scope, the following text was introduced:

Giannone et al. (2008) applied the Kalman filter, a rather complex technique to more than 200 time series, while we simply regress linearly GDP on about ten series. Naturally, our information set is smaller, so we may loose key data that could reduce the forecasting error further. Besides, our bridge models are simply confined to GDP as dependent variable, while Payne (2000), Baffigi et al. (2004) and Higgins (2014) have separate models for the expenditure components of GDP. So, we lost the possibility of having partial estimates, namely, for private consumption and investment, in order to keep things simple and flexible.